# DIVINE: Diverse-Inconspicuous Feature Learning to Mitigate Abridge Learning

## Abstract

Deep learning algorithms aim to minimize overall error and exhibit impressive performance on test datasets across various domains. However, they often struggle with out-of-distribution data samples. We posit that deep models primarily focus on capturing the prominent features beneficial for the task while neglecting other subtle yet discriminative features. This phenomenon is referred to as *Abridge Learning*. To address this issue and promote a more comprehensive learning process from data, we introduce a novel *DIVerse and INconspicuous feature lEarning* (DIVINE) approach aimed at counteracting Abridge Learning. DIVINE embodies a holistic learning methodology, effectively utilizing data by engaging with its diverse dominant features. Through experiments conducted on ten datasets, including MNIST, CIFAR10, CIFAR100, TinyImageNet, and their corrupted and perturbed counterparts (CIFAR10-C, CIFAR10-P, CIFAR100-C, CIFAR100-P, TinyImageNet-C, and TinyImageNet-P), we demonstrate that DIVINE encourages the learning of a rich set of features. This, in turn, boosts the model's robustness and its ability to generalize. The results on out-of-distribution datasets, such as those that are perturbed, achieve a performance 5.36%, 3.10%, and 21.85% mean Flip Rate (mFR) corresponding to CIFAR10-P, CIFAR100-P, and TinyImageNet-P datasets using DIVINE.On the other hand, Abridged Learning on CIFAR10-P, CIFAR100-P, and TinyImageNet-P datasets, achieve a performance 6.53%, 11.75%, and 31.90% mFR, respectively

## 1 Introduction

Deep learning algorithms have achieved tremendous success in several tasks including image classification (Dehghani et al., 2023), (Su et al., 2023), object detection (Wang et al., 2023b), (Zong et al., 2022), (Tan et al., 2020), and segmentation (Fang et al., 2023), (Xie et al., 2020). However, the robustness and generalizability of these algorithms in real-world scenarios is still an open problem. Supervised learning tasks in deep neural networks primarily focus on maximizing the classification accuracy by learning the easiest solutions/patterns that exist in the entire dataset (Geirhos et al., 2020). In other words, models take *shortcuts* by learning only the dominant input features that are sufficient for confident classification (Li et al., 2023),(Ilyas et al., 2019), (Pezeshki et al., 2021). This results in ignorance of other useful and distinct features that can be helpful in classification. Therefore, as shown in Figure 1, these models often fail to classify *out-of-distribution* samples.

In this research, we termed the above-mentioned learning process as "Abridge Learning". Formally, Abridge Learning (AL) is defined as the *"learning process in which a model learns only the dominant input features while failing to learn other useful input features for the target task".* The solution obtained by Abridge learning process lacks generalizability and is not suitable for deployment in real-world scenarios. To mitigate the problem of Abridge Learning, one solution is to identify the inconspicuous discriminative input features and use them to learn a diverse unified model that is generalizable to unseen real-world datasets. In an ideal training process, a model should identify and learn all the input features present; however, the identification of all the input features is intractable. Therefore, it is important to identify a set of diverse input features that can provide a generalized solution.

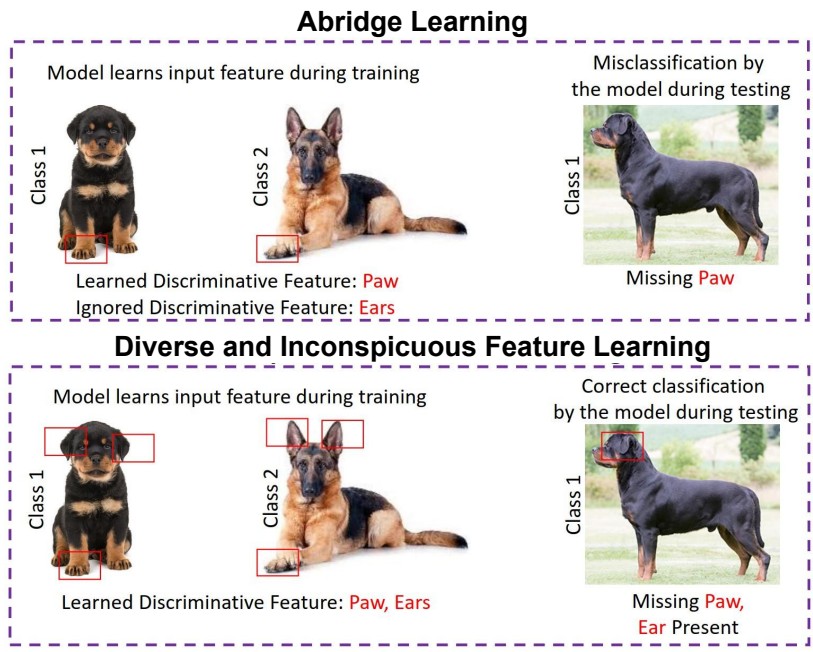

Figure 1: Illustration of the *Abridge Learning* (AL) and the proposed method. Model conventionally trained with AL, learns only 'paw' input feature and ignore other features. This results in failure of the model on the image with missing 'paw' feature. The proposed DIVINE learns 'paw' along with the feature 'ear' resulting in successful classification of the image where 'paw' feature is missing and the 'ear' feature is present.

**Research Contributions:** Existing methods address the problem of robustness and lack of generalizability in models; however they do not explicitly focus on learning the inconspicuous features from the dataset that are also discriminative in nature. This leads to learning of shortcut features that only work well for a given task on the given data but cannot sustain under real-world variations in data. To abridge this gap, we propose a novel learning method termed as DIVINE i.e., Diverse-Inconspicuous Feature Learning which mitigates this problem by removing the shortcuts and learning a diverse set of input features. The objective of the proposed method is two-fold: (a) identification of the minimum number of diverse inconspicuous discriminative input features as illustrated in Figure 1 and (b) train a unified model to learn the identified features for enhancing generalizability. The diverse features identified using the proposed method are disjoint, which in turn maximizes diverse learning of the model. Experiments for image classification tasks on CIFAR10, MNIST, CIFAR10-C, and CIFAR100-C datasets show that, DIVINE is generalized and can be applied to different machine learning tasks.

## 2 Related Work

The notion of "Abridge Learning" is excessively applicable to train deep learning models. In recent research, this learning schema has been termed as shortcut learning[1] (Geirhos et al., 2020) where the model finds shortcuts to minimize the loss, primarily by picking up only the dominant features in the input. This has been the traditional way of training the models. Carter et al. (2021) have shown that only 5% spurious pixel subsets are enough for confident prediction leading to over-interpretation of pixels resulting in shortcut learning. Such shortcuts were also recently identified in a medical imaging task (Oakden-Rayner et al., 2020) where the model picked undesired tumor patterns in the dataset. It gave a falsely reasonable performance, which went undetected because of the chosen evaluation metric. This implies that such spurious correlations can also remain unobserved because of the selected evaluation metrics. Lapuschkin et al. (2019) discussed Clever Hans strategies in the domain of Computer Vision and Arcade Gaming. They generated heatmaps and showed the patterns focused by the model while taking shortcuts.

Various drawbacks of shortcut learning process have recently been identified and addressed by the community. One such learning process termed as Gradient Starvation (Pezeshki et al., 2021) showed that the deep learning model extracts a subset of features to minimize the loss for training, while the gradients from other potentially essential features starve. Du et al. (2020) proposed a CREX method to avoid shortcuts. This method regularizes the subset of features annotated by experts. Wang et al. (2023a) proposed DFM-X, which use the prior knowledge from the previous models to train the target model to enhance the generalizability and robustness of the model. Zhang et al. (2023) proposed SADA method to generate images for the data augmentation corresponding to highly sensitive frequencies. Gao et al. (2023) proposed DDA based adaptation method, which learns the diffusion model on the source data and projects the target input to the source during testing. Guo et al. (2023) proposed a method to reduce the sensitivity of vision transformer against patch based corruptions. Ross & Doshi-Velez (2018) found that the networks trained with input gradients are more robust and generalizable. A similar approach to make models more robust involves regularizing the gradient norm of the model output with respect to the inputs. These Jacobian-based regularizers have been shown to significantly improve classification accuracy (Varga et al., 2017). Another approach incorporating the Jacobian in deep models is Jacobian Adversarially Regularized Networks (JARN), where the model's Jacobian is optimized by adversarial regularization (Chan et al., 2020). Similar insights have been utilized to evaluate the robustness of deep models on data containing random and adversarial input perturbations (Hoffman et al., 2019). The input perturbations and corruptions in the data are widely studied in the literature to evaluate the robustness of models (Rusak et al., 2020) (Taori et al., 2020). Different approaches (Bai et al., 2021; Strisciuglio & Azzopardi, 2022; Krueger et al., 2021; Benkert et al., 2022; Machireddy et al., 2022; Timpl et al., 2022; Chefer et al., 2022) are used to mitigate the effect of such distribution shifts. For example, Hendrycks et al. (2021) proposed a data augmentation method to address this problem, which includes geographic location as well as camera operation. Another work discusses the impact of manipulating batch normalization statistics for corrupted data to improve model performance (Schneider et al., 2020).

## 3  Diverse-Inconspicuous Feature Learning

As shown in Figure 2, the proposed DIVINE algorithm is a two-part learning approach. In the first learning part, the proposed algorithm identifies and then suppresses the learned input features via a dominance feature map to identify other inconspicuous input features. The dominance feature map represents the dominance of the identified input features for classification. Further, in the second part, a unified model is trained using these dominance feature maps to learn all the identified input features for enhancing generalizability. It should be noted that the problem of identification of all the inconspicuous features is intractable, and hence, the proposed method identifies a set of diverse inconspicuous features that maximize the learning of the model. As a result, the proposed algorithm is able to alleviate the problem of Abridged Learning, and the generalizability towards the out-of-distribution data samples.

Let $\mathbf{X} = \{(x_1, y_1), (x_2, y_2), ..., (x_n, y_n)\}$ be a dataset with original training images and their corresponding labels. For simplicity, let $x$ is an image of the dataset $\mathbf{X}$ with label $y$ in one-hot form. Consider a model $f_{\mathbf{X}}(.; \theta)$ with parameters $\theta$. This model is trained on the dataset $\mathbf{X}$ using cross entropy loss function, optimized over parameters $\theta$.

$$\min_{\theta} \ \mathbb{E}_{(x,y) \sim \mathbf{X}}[-y^T \log f_{\mathbf{x}}(x; \theta)] \tag{1}$$

where, $f_{\mathbf{X}}(x; \theta)$ outputs the probability vector for a given input image $x$. As mentioned earlier, the objective is to identify diverse inconspicuous input features via dominance feature maps followed by training a unified diverse model for enhancing generalizability. Here, we will first discuss the method for identification of inconspicuous features followed by training of the unified model.

---

[1]In the literature, Shortcut Learning is referred to as the broad term for various ways in which a model takes shortcuts by learning unintended strategies to minimize the loss. Abridge Learning is a sub-problem of Shortcut Learning that deals only with the problem where the model picks up only dominant cues and ignores other relevant features from the input data.

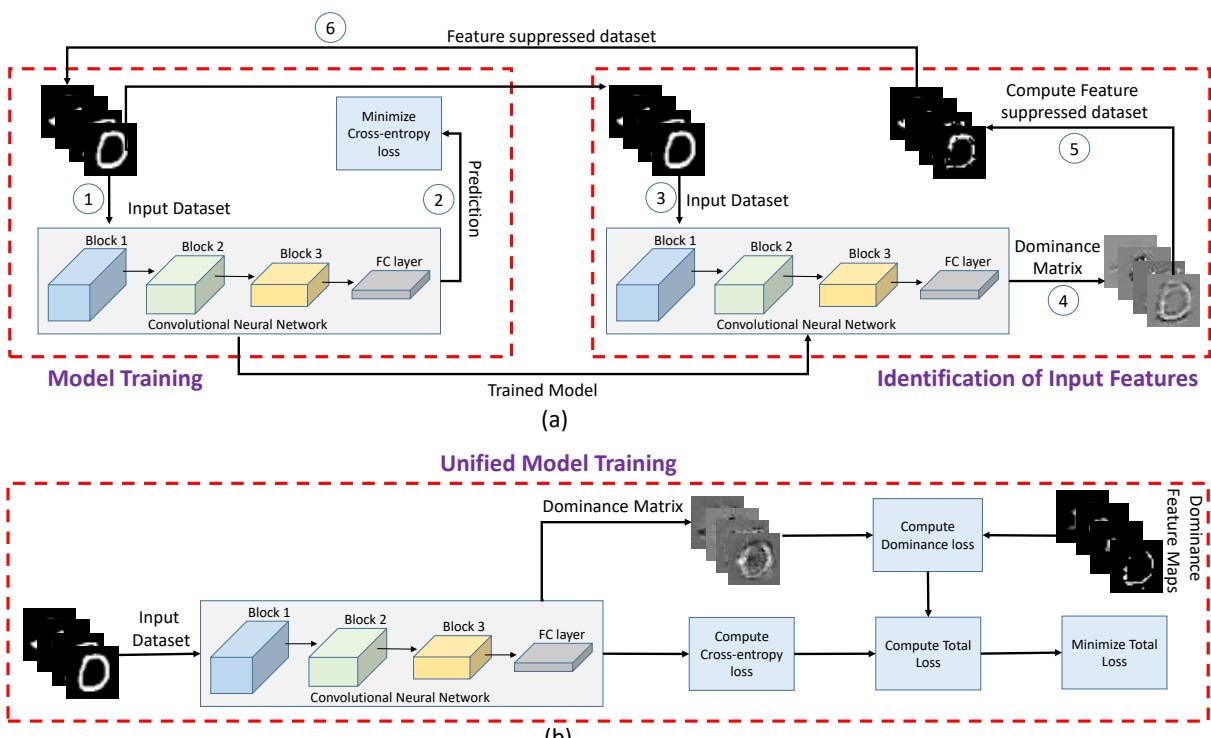

Figure 2: Pipeline of the proposed method for learning diverse and inconspicuous features. (a) Illustrates the process of identifying inconspicuous input features. Steps 1 and 2 involve training the model with original images from the dataset using a cross-entropy loss function. Steps 3 and 4 depict the computation of the dominance matrix for each image. In step 5, a feature-suppressed dataset is derived from the dominance matrix and dominance feature maps. The final step involves training the model with the feature-suppressed dataset to identify inconspicuous input features. (b) Demonstrates the unified model training process with original images.

## 3.1 Identification of Inconspicuous Input Features

Let $\mathbf{F}(\mathbf{x}) = \{F_1(x), F_2(x), ...F_r(x)\}$ be the set of all input features present in the image $x$ that can be learned by the model using the loss function in Eq. 1. Each feature $F_i(x)$ in the feature set $\mathbf{F}$ is the combination of input image pixels that can be learned by the model for classification. For example, in Figure 1, one of the image features (in the feature set learned by the model) constitutes of pixels of the *paw* region. From the literature (Geirhos et al., 2020) (Pezeshki et al., 2021), it is known that during training, a model learns the most dominant input features present in the dataset. Hence, the first step is to identify the dominance of each pixel in an input image $x$ for classification. For this purpose, we have used input-output Jacobian method (Chan et al., 2020) (Hoffman et al., 2019) that computes the dominance of each input image pixel on the model's output decision. Mathematically, for a given image $x$ with input perturbation $\epsilon$, the Taylor series expansion of the function $f(x + \epsilon; \theta)$ is defined as:

$$f(x + \epsilon; \theta) = f(x; \theta) + \epsilon \frac{df(x; \theta)}{dx} + \mathcal{O}(\epsilon^2) \tag{2}$$

The higher-order terms can be neglected for very small perturbations. Hence, the above equation is updated as:

$$f(x + \epsilon; \theta) \approx f(x; \theta) + \epsilon \frac{df(x; \theta)}{dx} \tag{3}$$

where, term $\frac{df(x;\theta)}{dx}$ represents the input-output Jacobian matrix. Since we are computing Jacobians with respect to the true class only, we term the output matrix as Dominance matrix denoted by $D_1(x) = \frac{df(x;\theta)}{dx}$. Large values (both +ve and -ve) in the dominance matrix represent higher dominance of the corresponding image pixels in the input image $x$. In other words, the model's decision is highly dependent on the input image pixels with high dominance values in the dominance matrix.

Once we obtain the dominance matrix, the next step is to identify the image pixels with higher dominance values. Let $p$ be the percentage of the most dominant image pixels. The combination of these identified pixels represents the first identified input feature $F_1(x)$. To obtain $F_1(x)$, a mask $M_1(x)$ is created using the following function:

$$M_1(x) = \begin{cases} 1 & \text{if} \quad |D_1(x)| \geq t \\ 0 & \quad \text{otherwise} \end{cases} \tag{4}$$

Here, $t$ is the threshold obtained by sorting the dominance values of the dominance matrix for top $p$ percentage of pixels. Next, we compute the element-wise multiplication of the mask with the input image $x$ for obtaining feature $F_1(x)$. Mathematically, it is written as:

$$F_1(x) = M_1(x) \odot x \tag{5}$$

After obtaining the feature $F_1(x)$, the next step is to compute the dominance feature map $D_{m_1}(x)$ corresponding to the identified feature $F_1(x)$. These maps are used during unified model training. Here, dominance feature map $D_{m_1}(x)$ represents the dominance values of identified feature $F_1(x)$ i.e., the combination of identified dominant pixels. The dominance map $D_{m_1}(x)$ is obtained by:

$$D_{m_1}(x) = M_1(x) \odot D_1(x) \tag{6}$$

Next, we identify other diverse inconspicuous input features. Conceptually, diversity can be achieved by identifying features that are completely different from one another. Thus, we enforced identified features to be disjoint, i.e., $F_i(x) \cap F_j(x) = \phi$, for $i \neq j$. For this purpose, we have suppressed the identified input feature $F_1(x)$ in the input image $x$. This is done by setting the input image pixels to zero corresponding to the non-zero dominance values in the dominance feature map $D_{m_1}(x)$. Mathematically, it is written as:

$$x_{s_1} = \begin{cases} x & \text{if} \quad |D_{m_1}(x)| = 0 \\ 0 & \text{if} \quad |D_{m_1}(x)| \neq 0 \end{cases} \tag{7}$$

where, $x_{s_1}$ is the output image with suppressed feature $F_1(x)$. The above-mentioned process is applied to all the training images in the dataset $\mathbf{X}$. This results in a new dataset $\mathbf{X}_{s_1}$ with suppressed features obtained corresponding to all the images. It is important to note that suppressing the identified features corresponding to all the images in the dataset $\mathbf{X}$ will enforce the model to learn other features.

To identify other input features, we use the feature-suppressed dataset $\mathbf{X}_{s_1}$. For this purpose, we have trained a separate model using the loss function mentioned in Equation 1. Then, the process from Equations 2 to 7 is repeated using $\mathbf{X}_{s_1}$ in place of $\mathbf{X}$ to obtain $\mathbf{X}_{s_2}$. This process is repeated to identify input features until the stopping criteria is achieved. Figure 3 shows samples of the original and feature-suppressed datasets corresponding to MNIST dataset. The details of the stopping criteria are discussed in Subsection 3.6.

## 3.2 Diverse Unified Model Learning

Once the input features are identified, the next objective is to train a unified model that learns all the identified input features. For this purpose, we have used the dominance feature maps corresponding to all the identified features. Let $k$ be the number of input features identified for each image $x$. This means we have $k$ number of dominance feature maps i.e., $D_{m_1}(x), D_{m_2}(x), ..., D_{m_k}(x)$. To provide supervision during training, all the dominance feature maps are combined into a single unified map by adding them together. Mathematically, it is written as:

$$D_m(x) = D_{m_1}(x) + D_{m_2}(x) + ... + D_{m_k}(x) \tag{8}$$

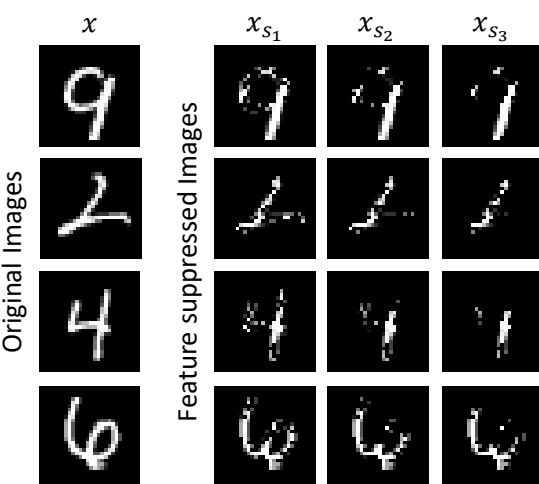

Figure 3: Sample of original images and the corresponding intermediate feature-suppressed images from the MNIST dataset obtained using the proposed method.

where, $D_m(x)$ is the unified dominance feature map. Let $f_u(.; \theta)$ be the unified model with parameters $\theta$ to be trained on the dataset $\mathbf{X}$ with original images. In order to enforce the model to learn all the identified input features, the following objective function is used.

$$\min_{\theta} \ \mathbb{E}_{(x,y) \sim \mathbf{X}}[-y^T \log f_u(x; \theta) + L_D(x)] \tag{9}$$

Here, the first term represents the standard cross-entropy loss while the second term $L_D(x)$ represents the dominance loss. The dominance loss enforces the model to focus on the identified input features. Let $D_u(x)$ be the Dominance matrix with dominance values corresponding to the unified model $f_u(x; \theta)$. Then, in order to compute the dominance loss, the squared Euclidean distance between the unified dominance feature map $D_m(x)$ and the dominance matrix $D_u(x)$ is minimized. Mathematically, the loss $L_D(x)$ is written as:

$$L_D(x) = ||D_m(x) - D_u(x)||_2^2 \tag{10}$$

Therefore, the final objective function is written as :

$$\min_{\theta} \ \mathbb{E}_{(x,y) \sim \mathbf{X}}[-y^T \log f_u(x; \theta) + ||D_m(x) - D_u(x)||_2^2] \tag{11}$$

Since the unified model learns inconspicuous and diverse input features, thereby, it not only alleviates the effect of Abridged Learning but also enables the model to generalize over out-of-distribution samples.

### 3.3 Experimental Setup

The primary objective of this paper is to remove the *shortcuts* learned by the model via learning inconspicuous and diverse input features. Our hypothesis is based on the observation that existing algorithms learn dominant features while ignoring other relevant features from the dataset within a distribution. The performance of the models suffers when dominant features are distorted/suppressed. This is because the inductive bias of the model is based on the dominant feature only. To address this, the proposed DIVINE algorithm is designed to learn the dominant features along with inconspicuous features reducing the dependence on the dominant feature only. Since the proposed learning process introduces the model to suppressed features as well, making the training diverse in nature. This ensures the generalized inductive bias of the final trained model. In order to validate this hypothesis, the experiments are performed for Abridged Learning on the feature-suppressed datasets corresponding to MNIST (LeCun et al., 1998), CIFAR10 (Krizhevsky et al., 2009), and TinyImageNet (Le & Yang, 2015) datasets. These feature-suppressed datasets are obtained by suppressing the identified input features (described in Section 3.1).

Table 1: Comparison of classification accuracy (%) of existing algorithms and the DIVINE on the original and feature-suppressed datasets.

| | Abridge Learning | Jacobian Regularization | Random Suppression | DIVINE |
|---|---|---|---|---|
| **MNIST** | | | | |
| **Original** | 99.21 | 85.80 | 98.67 | 97.31 |
| $\mathbf{X}_{s_1}$ | 66.53 | 76.52 | 85.56 | 91.41 |
| $\mathbf{X}_{s_2}$ | 50.98 | 70.18 | 68.10 | 84.98 |
| $\mathbf{X}_{s_3}$ | 40.87 | 62.97 | 54.24 | 75.20 |
| **Average** | 64.40 | 73.87 | 71.64 | 87.22 |
| **CIFAR10** | | | | |
| **Original** | 82.38 | 81.63 | 81.73 | 80.34 |
| $\mathbf{X}_{s_1}$ | 64.16 | 65.90 | 63.97 | 67.41 |
| $\mathbf{X}_{s_2}$ | 52.63 | 54.88 | 52.01 | 58.18 |
| $\mathbf{X}_{s_3}$ | 44.45 | 46.87 | 45.19 | 51.83 |
| **Average** | 60.98 | 62.32 | 60.72 | 64.44 |
| **CIFAR100** | | | | |
| **Original** | 74.63 | 75.95 | 75.46 | 74.57 |
| $\mathbf{X}_{s_1}$ | 37.44 | 48.31 | 45.21 | 56.31 |
| $\mathbf{X}_{s_2}$ | 27.84 | 39.61 | 36.86 | 49.79 |
| $\mathbf{X}_{s_3}$ | 15.22 | 26.28 | 14.71 | 33.49 |
| **Average** | 39.23 | 44.64 | 43.06 | 53.54 |
| **Tiny-ImageNet** | | | | |
| **Original** | 60.59 | 49.06 | 51.66 | 54.59 |
| $\mathbf{X}_{s_1}$ | 40.46 | 39.29 | 43.71 | 43.42 |
| $\mathbf{X}_{s_2}$ | 27.35 | 31.91 | 33.92 | 32.67 |
| $\mathbf{X}_{s_3}$ | 21.67 | 27.28 | 27.94 | 26.66 |
| **Average** | 37.51 | 36.88 | 39.30 | 39.33 |

To further analyze the applicability of the proposed algorithm in real-world scenarios, we perform the experiment by evaluating the unified model on out-of-distribution samples. These samples are taken from the corrupted datasets CIFAR10-C, CIFAR100-C, TinyImageNet-C and perturbed datasets CIFAR10-P, CIFAR100-P, and TinyImageNet-P. The corrupted datasets contain 15 different corruptions corresponding to the CIFAR10, CIFAR100, and TinyImageNet datasets, respectively. We discuss the details of the corruption and perturbed datasets employed for evaluation in supplementary.

### 3.4 Existing Approaches for Comparison

Since the proposed method involves the suppression of different input pixels under the supervision of dominance maps, we have performed random suppression of input features for comparison. In the random suppression method, we randomly drop $p\%$ pixels during training. Further, we have compared our method with Jacobian Regularization. In literature, Jacobian Regularization (Chan et al., 2020) is primarily used for reducing the sensitivity of input image pixels to enhance the robustness of the models. We have used this regularization method with the standard cross-entropy loss function.

### 3.5 Evaluation Metrics

We compute the classification accuracy to evaluate model performance on the MNIST and CIFAR10 datasets. To evaluate the performance on the corrupted images, we report the Relative Corruption Error (Relative CE) and Relative Mean Corruption Error (Relative mCE) (Hendrycks & Dietterich, 2018).

$$\text{Relative } CE_c^f = \frac{\sum_{s=1}^{5} E_{s,c}^f - E_{original}^f}{\sum_{s=1}^{5} E_{s,c}^b - E_{original}^b} \tag{12}$$

where, $f$ denotes the model to be evaluated, $b$ denotes the baseline model obtained using Abridge Learning (AL), $E_{clean}^f$ and $E_{clean}^b$ denote the error obtained corresponding to the model to be evaluated and the AL model on original images. $E_{s,c}^f$ and $E_{s,c}^b$ denote error rates on corruption $c$ at severity level $s$ corresponding

Table 2: Comparison of relative corruption error obtained using existing and proposed methods for each corruption type on the CIFAR10-C and CIFAR100-C datasets.

| Corruptions | CIFAR10-C | | | CIFAR100-C | | |
|---|---|---|---|---|---|---|
| | Jacobian Regularization | Random Suppression | DIVINE | Jacobian Regularization | Random Suppression | DIVINE |
| Defocus Blur | 102.35 | 108.77 | 92.90 | 105.88 | 108.38 | 102.08 |
| Contrast | 100.19 | 103.55 | 99.64 | 103.86 | 114.32 | 122.66 |
| Pixelate | 95.09 | 113.08 | 75.58 | 108.59 | 106.95 | 74.61 |
| Snow | 91.57 | 81.15 | 81.38 | 97.17 | 99.32 | 78.91 |
| Fog | 105.04 | 111.62 | 99.17 | 111.01 | 110.61 | 110.52 |
| Glass Blur | 96.11 | 87.62 | 73.69 | 98.51 | 98.49 | 82.99 |
| Brightness | 106.40 | 133.70 | 101.00 | 129.75 | 136.48 | 125.60 |
| Elastic | 103.84 | 109.48 | 97.95 | 107.35 | 106.55 | 96.89 |
| Frost | 100.47 | 113.05 | 88.12 | 104.53 | 111.80 | 83.74 |
| JPEG | 86.40 | 93.95 | 70.86 | 93.23 | 91.82 | 73.24 |
| Shot Noise | 95.65 | 111.77 | 72.62 | 101.47 | 107.27 | 76.46 |
| Impulse Noise | 91.70 | 92.17 | 73.02 | 105.16 | 96.99 | 78.15 |
| Zoom Blur | 99.16 | 111.06 | 95.85 | 97.47 | 98.42 | 89.50 |
| Gaussian Noise | 95.49 | 110.70 | 73.32 | 101.74 | 107.97 | 79.18 |
| Motion Blur | 105.37 | 116.31 | 99.40 | 107.86 | 113.26 | 98.38 |
| **Relative mCE** | 98.32 | 106.29 | **86.30** | 104.90 | 107.24 | **91.53** |

to the model to be evaluated and the AL model, respectively. A lower Relative CE indicates a higher performance over the baseline.

To evaluate the performance on the perturbed datasets, we measure the probability that two consecutive frames with different intensity of perturbations, have "flipped" or mismatched predictions. This is termed as mean Flip Rate (mFR) Hendrycks et al. (2019).

### 3.6 Selection Criteria for Number of Feature Maps

In order to decide the number of features, we have computed the running average of the classification accuracy obtained on the original and feature suppressed dataset using AL method. We decided to iterate computing feature maps at most 3 times given the average running classification should not be below 50 percent of the classification accuracy obtained on the original dataset.

## 4 Results and Analysis

The proposed DIVINE algorithm is evaluated on two types of datasets: (i) feature-suppressed datasets and (ii) corruptions. In the first set of evaluations, we validate our assertion of "Abridge Learning" using the MNIST, CIFAR10, CIFAR100, and TinyImageNet datasets. For corruptions, we showcase results on CIFAR10-C, CIFAR100-C, and TinyImageNet-C datasets. We further compare the DIVINE algorithm with a recent algorithm proposed by Carter et al. (2021).

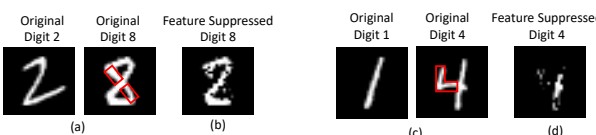

Figure 4: Visualizations of the semantically relevant features learned by the model. (a) shows the strokes learned by the model, which distinguishes digit 8 from 2, and (b) shows the feature-suppressed image. Similarly, (c) and (d) show the distinguishing strokes learned by the model and the feature-suppressed image.

Table 3: Classification accuracy (%) obtained using Jacobian Regularization (JR), Random Suppression (RS), and DIVINE algorithm on the TinyImageNet-C dataset for different corruptions.

| Corruptions | Jacobian Regularization | Random Supression | DIVINE |
|---|---|---|---|
| Defocus Blur | 21.07 | 19.97 | 29.43 |
| Contrast | 12.37 | 12.51 | 18.70 |
| Pixelate | 27.98 | 31.79 | 39.53 |
| Snow | 25.16 | 26.83 | 32.8 |
| Fog | 21.46 | 23.37 | 33.09 |
| Glass Blur | 28.33 | 30.36 | 32.14 |
| Brightness | 27.02 | 28.33 | 36.14 |
| Elastic | 21.23 | 21.47 | 30.89 |
| JPEG | 26.30 | 29.07 | 37.25 |
| Shot Noise | 35.77 | 37.68 | 43.09 |
| Impluse Noise | 29.71 | 34.09 | 39.14 |
| Zoom Blur | 20.07 | 18.83 | 28.20 |
| Gaussian Noise | 34.75 | 36.99 | 41.84 |
| Motion Blur | 21.47 | 22.11 | 30.87 |
| **Mean** | 25.19 | 26.67 | **33.79** |

## 4.1 Evaluation on Feature-suppressed Datasets

This experiment is performed to validate that a model trained using conventional methods relies on the dominant features (which are easy to learn) during classification, thereby reducing the performance of the model when these dominant input features are missing. The performance of the AL model trained on original images is evaluated on the testing set of original and feature-suppressed datasets, i.e., $\mathbf{X}$, $\mathbf{X}_{s_1}$, $\mathbf{X}_{s_2}$ and $\mathbf{X}_{s_3}$. Dataset $\mathbf{X}_{s_1}$ has images with one suppressed dominant input feature in each image. Similarly, datasets $\mathbf{X}_{s_2}$ and $\mathbf{X}_{s_3}$ have images with two and three suppressed dominant input features in each image, respectively. Table 1 shows the performance of the AL models corresponding to the MNIST, CIFAR10, CIFAR100, and TinyImageNet datasets. It is observed that the performance of the AL models degrades significantly on the feature-suppressed datasets. For instance, the performance of the AL model trained on the MNIST dataset drops from 99.21% to 66.53% on feature-suppressed dataset $\mathbf{X}_{s_1}$ (32.68% drop), which further degrades to 50.98% on feature-suppressed dataset $\mathbf{X}_{s_2}$. This shows that the performance of the models trained using conventional methods is heavily dependent on the dominant input features. On the other hand, the performance of the unified model drops by only 5.90% when evaluated on $\mathbf{X}_{s_1}$. As seen in Table 1, the unified model trained using the proposed DIVINE algorithm performs well on the feature-suppressed datasets. In Figure 5, the first row depicts the feature-suppressed images obtained for a sample of class 'horse'. From $x_{s_1}$, it is apparent that the model is focusing on the pixels associated with the body of the horse (dominant features). However, after two iterations, the model starts focusing on other (inconspicuous) features such as the tail of the horse. This highlights that the DIVINE algorithm learns both dominant and inconspicuous

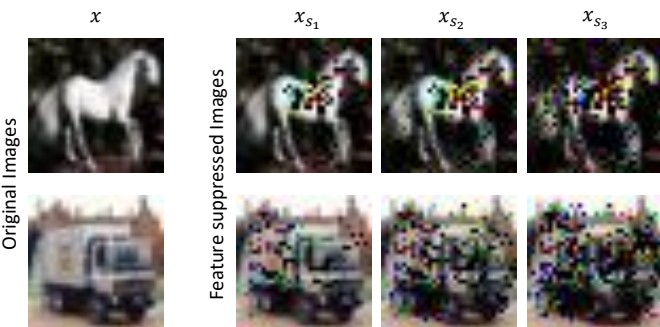

Figure 5: A few sample original images and the corresponding intermediate feature-suppressed images on the CIFAR10 dataset obtained using the proposed method.

Table 4: Classification accuracy (%) obtained using Abridged Learning and DIVINE algorithm on the CIFAR10-P and CIFAR100-P datasets for different perturbations.

| mFR % | CIFAR10 | | CIFAR100 | |
|---|---|---|---|---|
| | Abridged Learning | DIVINE | Abridged Learning | DIVINE |
| Brightness | 1.33 | 1.22 | 2.98 | 1.03 |
| Gaussian Noise | 5.21 | 3.86 | 15.3 | 2.21 |
| Motion Blur | 11.26 | 9.9 | 14.74 | 2.1 |
| Rotate | 8.29 | 6.24 | 11.12 | 3.09 |
| Scale | 9.55 | 7.99 | 13.2 | 4.87 |
| Shot Noise | 6.4 | 4.82 | 17.96 | 2.9 |
| Snow | 3.75 | 2.96 | 6.59 | 1.15 |
| Tilt | 3.13 | 2.49 | 5.52 | 1.55 |
| Translate | 15.63 | 13.53 | 28.34 | 11.79 |
| Zoom Blur | 0.79 | 0.67 | 1.78 | 0.32 |
| Overall mFR | 6.534 | **5.368** | 11.753 | **3.101** |

features, not depending only on the dominant features for classification. It is also observed that the models learn semantically relevant features in the feature-suppressed datasets. Figure 4 (a) & (b), show that the discriminative stroke of digit '8' (highlighted in red) is the most dominant feature differentiating it from digit '2', and is therefore suppressed. Similar observations can be made for digits '4' and '1' in Figure 4 ((c) & (d)).

The visualization of the dominance matrix computed corresponding to both the AL and unified models is shown in Figure 6. It is observed that the dominance matrix computed corresponding to the AL models and the models trained using feature-suppressed datasets are focused on specific input features. On the other hand, the dominance matrix of the unified model is focused on all identified input features. This results in the high performance of the unified model on the feature-suppressed datasets. Results of the unified model are compared with random suppression, and the Jacobian regularization method (Chan et al., 2020). Both approaches are used to enhance the robustness of the models. It is observed that existing approaches perform better than the AL model, especially on the MNIST dataset. However, the performance is not as good as that obtained using the unified model. In the random suppression method, there is no supervision to the model for learning a diverse set of features. While in the Jacobian regularization method, the model reduces its dependency on the dominant features during training, it is not able to learn the inconspicuous features.

**Ablation Study for parameter** $p$**:** On updating the values of $p$ from 3% to 10%, the performance of the unified model degrades on feature-suppressed datasets. Since, the model prediction is dependent only on a few input pixels, setting a higher value of $p$ results in suppressing of dominant as well as other input features, which in turn decreases the performance of the unified model on the feature-suppressed datasets.

## 4.2 Evaluation on Corruptions and Perturbations

This experiment is performed to evaluate the generalizability and robustness of the proposed unified model on out-of-distribution images. The performance on corruptions and perturbations are discussed below:

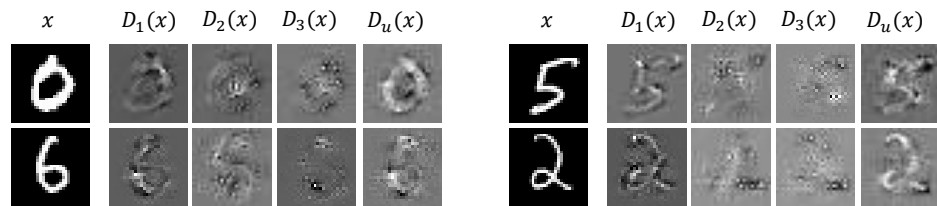

Figure 6: Illustration of the dominant features and inconspicuous features obtained in the MNIST dataset. Sample original images $x$ and the corresponding dominance matrices $D_1(x)$, $D_2(x)$, $D_3(x)$, and $D_u(x)$. $D_1(x)$ and $D_u(x)$ are obtained corresponding to the AL and unified model trained on the original dataset. In contrast, $D_2(x)$ and $D_3(x)$ are obtained corresponding to the model trained on feature-suppressed datasets.

Table 5: Classification accuracy (%) obtained using Abridged Learning and DIVINE algorithm on the TinyImageNet-P dataset for different perturbations.

| mFR % | TinyImageNet | |
|---|---|---|
| | Abridged Learning | Unified Model |
| Brightness | 12.09 | 9.53 |
| Gaussian Noise | 28.56 | 18.58 |
| Gaussian Noise V3 | 56.82 | 42.36 |
| Rotate | 45.49 | 30.19 |
| Shear | 37.83 | 24.19 |
| Shot Noise V2 | 41.11 | 26.99 |
| Snow | 15.66 | 11.26 |
| Specekle Noise | 26.4 | 15.26 |
| Specekle Noise V3 | 53.97 | 38.08 |
| Translate | 41.92 | 31.51 |
| Gaussian Blur | 5.17 | 3.91 |
| Gaussian Noise V2 | 47.06 | 32.98 |
| Motion blur | 7.66 | 5.36 |
| Scale | 40.3 | 29.87 |
| Shot Noise | 32.09 | 20.14 |
| Shot Noise V3 | 50.29 | 34.85 |
| Spatter | 17.5 | 12.22 |
| Specekle Noise V2 | 44.1 | 28.65 |
| Tilt | 25.26 | 15.45 |
| Zoom Blur | 8.73 | 5.74 |
| Overall mFR | 31.9005 | **21.856** |

**Performance on Corruption**   We have reported the relative corruption error (Relative CE) and the relative mean corruption error (Relative mCE) on the CIFAR10-C and CIFAR100-C datasets. The results are shown in Table 2. The proposed unified model outperforms random suppression and Jacobian Regularization methods on the corruption datasets. The proposed unified model gives a Relative mCE of **86.30** and **91.53** corresponding to the CIFAR10-C and CIFAR100-C datasets, respectively. On comparing the performance of CIFAR10-C and CIFAR100-C datasets for individual corruptions, the proposed unified model trained with the DIVINE method outperforms other methods on 14 corruptions (excluding snow corruption on the CIFAR10-C and contrast corruption on the CIFAR100-C) and gives a comparable performance on the snow and contrast corruption corresponding to CIFAR10-C dataset, respectively. Table 1 shows that the performance of all the existing methods is comparable to the original images. However, random suppression and Jacobian regularization methods fail to generalize well on out-of-distribution images. This happens because existing approaches focus on learning dominant features only, which may be absent/distorted in the corrupted images. The proposed model performs better as it has learned diverse and inconspicuous features which are helpful for classification.

To test our method on a large-scale dataset, we computed the performance on the TinyImageNet-C dataset and report the results obtained in Table 3. On the TinyImageNet-C dataset, DIVINE yields an absolute improvement of 25.54% and 19.02% over the random suppression and Jacobian regularization methods, respectively as shown in Table 3. We observe the robustness of the proposed DIVINE algorithm against a variety of corruptions. These results are illustrated for three different learning methods namely- Jacobian Regularization, Random Suppression, and the proposed DIVINE algorithm. From the table, it is clearly visible that the proposed algorithm outperforms other algorithms on all corruptions. We also achieve significantly better mean classification accuracy using the proposed DIVINE algorithm. This clearly describes the applicability of DIVINE algorithm on large-scale datasets as well.

**Performance on Perturbations**   We have reported the Flip Rate (FR) and the Overall mean flip rate (Overall mFR) on the CIFAR10-P, CIFAR100-P, and TineImageNet-P datasets. The results are shown in Table 4 and 5. The proposed unified model outperforms abridge learning on the perturbed datasets. The proposed unified model gives an overall mFR of **5.36%**, **3.10%**, and **21.85%** corresponding to the CIFAR10-P, CIFAR100-P, and TinyImageNet-P datasets, respectively. On comparing the performance of individual

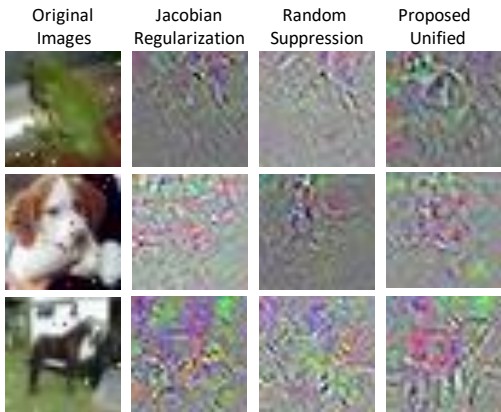

Figure 7: Sample images corrupted with impulse noise and the corresponding dominance matrices obtained using Jacobian regularization, random suppression, and the proposed unified model (Best viewed in color).

perturbations on all datasets, the proposed unified model trained with the DIVINE method outperforms abridge learning.

The visualization of the dominance matrix obtained corresponding to the unified model and existing approaches on the corrupted images of the CIFAR10 dataset are shown in Figure 7. We can see that the unified model focuses on multiple input features/regions while the existing approaches fail to focus diversely. On observing the relative corruption error corresponding to impulse noise corruption in Table 2, it is found that the error for the proposed method is 73.02, which is 18.68 and 19.15 less than Jacobian regularization and random suppression methods, respectively. This shows the applicability of the proposed method in real-world scenarios where external corruptions are common. Additionally, the proposed method can be used in combination with existing approaches for improving robustness.

**Comparison with Carter et al. (NeurIPS 2021) (Carter et al., 2021):** Carter et al. Carter et al. (2021) have shown that only 5% spurious pixel subsets in an image are enough for confident prediction. These pixel subsets may be meaningless to humans and lead to over-interpretation by the model. This validates our assumption of abridge learning during training of the model. The authors further used an ensemble method to mitigate the problem of abridge learning. In order to showcase the effectiveness of the proposed DIVINE algorithm, we have also compared its performance on CIFAR10-C dataset. The DIVINE algorithm achieves a relative mCE of 86.30 which outperforms the ensemble method (Carter et al., 2021) by a margin of 8.62. In general, ensemble methods reduce overfitting and improve model performance. However, they do not necessarily make the model robust towards out-of-distribution samples. By enforcing the learning of inconspicuous input features, the DIVINE algorithm offers better robustness.

### 4.3 Ablation Experiment to Visualize the Trend of Parameter $p$

We conduct a series of experiments for varying values of $p$, specifically at 0.5%, 1.5%, 3%, 6%, and 10%. The trend observed in average classification accuracy for these values is depicted in Figure 8, and detailed results are presented in Table 1. From this representation, we notice a slight decrease in accuracy at $p$=0.5% and

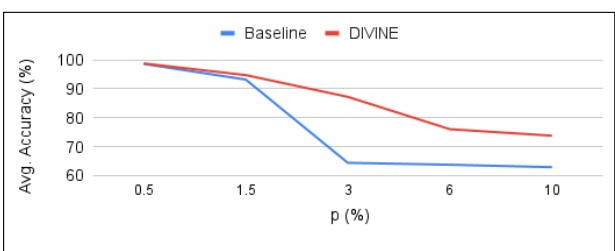

Figure 8: Plot representing the trend line of the average classification accuracy corresponding to baseline and proposed algorithm on different values of p on the MNIST dataset.

Table 6: Training time of Abridged Learning and proposed DIVINE method on CIFAR10 dataset.

| | Abridged Learning | DIVINE |
|---|---|---|
| Time per Epoch (in seconds) | 73 | 177 |
| Total Training Time (in minutes) | 25 | 60 |

$p$=1.5% for the baseline model, indicating that dominant features are still present in the datasets with feature suppression at these lower $p$ values. However, at $p$=3%, there is a noticeable decline in the baseline model's performance, highlighting the successful elimination of dominant features in the dataset. Consequently, we have selected $p$=3% as the optimal value for conducting our experiments.

### 4.4 Computational Runtime

We have calculated the temporal cost associated with the MNIST, CIFAR10, CIFAR100, and TinyImageNet datasets during the first phase. In this phase, the model undergoes training on the dataset with suppressed features, and this process is repeated three times. The MNIST dataset requires 10 epochs for training, with each epoch taking approximately 14 seconds to complete. The entire training process for MNIST, therefore, takes around 420 seconds, equivalent to 7 minutes. In the case of CIFAR10 and CIFAR100 datasets, each epoch lasts about 61 seconds, while for the TinyImageNet dataset, it is around 935 seconds per epoch. These time durations present opportunities for optimization, possibly through the application of methods like the one proposed by Wang et al. (Wang et al., 2020), which involves pruning the network at the initial stage, prior to training. It is important to note that the time required for inference (testing) remains consistent between the proposed unified model and the AL model. Further, we compute the time complexity of Abridge Learning and the proposed DIVINE algorithm for each epoch as well as the total training time. From Table 6 we oberve that overhead training time is 104 seconds for each epoch, which is mostly spent on the computation of Jacobians. Since, the proposed DIVINE method increase the computation time over Abridge Learning, we consider this a limitation of the proposed method the minimization of this overhead can be explored in the future work.

## 5 Limitations

We highlight the following limitations of the proposed DIVINE algorithm:

- Though the proposed method mitigates Abridged Learning, it is computationally expensive as it utilizes multiple iterations to identify inconspicuous and diverse features and training of the models on the same data.

- The proposed idea is validated only for the image modality in classification setting. However, the proposed approach can be extended for more modalities and different tasks.

- The proposed method DIVINE is not applicable to those abridge learning problems where the dominant features are not semantic. For example, in ColoredMNIST dataset, the spurious correlation is due to color attribute of the digits, which is not a semantic feature and DIVINE is capable of only highlighting the semantic features in the input data.

## 6 Conclusion

Conventional deep learning algorithms typically prioritize enhancing classification accuracy, which can result in inadequate learning and poor generalization to out-of-distribution samples. In this paper, we introduce a unique, holistic learning approach called *Diverse and Inconspicuous Learning* (DIVINE) which focuses on maximizing learning from a given set of inputs and learning as many discriminative features as possible. We validate DIVINE's effectiveness through extensive experiments across various datasets, including MNIST, CIFAR10, CIFAR10-C, CIFAR10-P, CIFAR100-C, CIFAR100-P, TinyImageNet, TinyImageNet-C, and TinyImageNet-P. The results reveal that the dominance maps generated via our method provide superior guidance for learning a rich set of input features. Consequently, our model demonstrates enhanced generalizability and robustness, particularly in the presence of out-of-distribution samples. We posit that this

comprehensive style of learning ensures more reliable model predictions, especially in real-world situations where data corruption and distribution shifts can significantly impair performance.

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
