# OpenReview forum: "DIVINE: Diverse-Inconspicuous Feature Learning to Mitigate Abridge Learning"
_TMLR — Rejected by TMLR_

### Review · Reviewer_JB4q · 2023-11-12

**Summary Of Contributions:**

This paper proposes Abridge Learning to alleviate the effect that the feature extractor in DNN tends to overlook a part of discriminative features (this is also called shortcut learning). To mitigate this challenge, DIVINE is proposed to counteract Abridge Learning. Experiments on some standard benchmarks show the proposed method works well and can boost the model's robustness.

**Audience:**

Yes

**Claims And Evidence:**

Yes

**Requested Changes:**

Please refer to Weaknesses.

**Strengths And Weaknesses:**

Strengths:
1. This paper is well-written, it is enjoyable to read.
2. The core idea in DIVINE is intuitive and interesting.
3. The targeted problem is valuable and important for deep learning algorithms.

Weaknesses:
1. Feature learning is a popular topic in deep learning theory recently, while feature learning is a main contribution in this paper. Is there any connection between the proposed method with theory?
2. The proposed approach directly sets the input image pixels to zero corresponding to the non-zero dominance values in the dominance feature matrix, which is easy but somewhat unreasonable.
3. The learning process is called "Abridge Learning". It is not very clear for me why the new learning setup should be newly defined.
4. It is also not clear how to select the number of identified feature maps.
5. In Eq.8, why choosing use the operation of "sum" to fusion the identified feature maps?
6. The experiments results in Table 1 are not very convincing. The proposed DIVINE directly change the value of pixel in pixel space. It is not very reasonable to verify the model's performance in feature-suppressed datasets.

---

> ### Author Response · Authors · 2023-12-10
> **Detailed response to comments of Reviewer JB4q**
>
> We thank the reviewer for providing their feedback and suggestions. Our responses to the reviewer’s questions are provided below:
>
> - **Discussion on feature learning and DIVINE method:** In the literature, the feature learning algorithms are primarily focused on using (1) model space by making changes in existing model architecture or proposing new model architecture and (2) feature space by applying loss functions in feature space to achieve the desired task. In the proposed method DIVINE, we are using the image space by identifying the combinations of pixels that represent the features learned by the model and further learning the ignored features in the feature space. Learning and identifying features in image space provides better explainability of the model behavior.
>
> - **Why we set the pixel value to 0 for suppressing the feature?** Setting the pixel value to 0 is one way of suppressing the leaned feature. In general, we consider the presence of black pixels/regions as the absence of the feature. For example, in the MNIST database, the background is black, which we consider the absence of the feature. We agree that there could be other ways to suppress the identified feature. However, we chose to set the pixel value to zero as it is simple and more interpretable to humans as well.
>
> - **Clarity on the learning process “Abridge Learning”:** As mentioned in the paper, Abridge learning is a learning process in which a model takes the shortcut strategies to learn only the dominant input features while ignoring other useful input features for the target task. In other words, the model only learns the subset of relevant features for the given task and fails to learn the other important features. For example, consider a dataset having images of moon and star patterns, where the moon in the images is situated at the top left and the stars in the images are situated at the bottom right. Now, the task is to classify the moon vs star images. As shown in the literature [1], the deep models learn to associate object location with a category, which represents abridge learning.
>
> - **Number of feature maps:** In order to decide the number of features, we have computed the running average of the classification accuracy obtained on the original and feature-suppressed dataset using the AL method. We decided to iterate computing feature maps at most 3 times, given the average running classification should not be below 50 percent of the classification accuracy obtained on the original dataset. We have included the computation of feature maps in section 3.6 in the revised manuscript.
>
> - **Fusion of Feature maps in equation 8:** In the DIVINE method, we are iteratively identifying the features learned by the model and suppressing the same to enforce the model to learn other features. We are setting the pixel values to zero in every iteration of the identified features for the suppression of the features. It enforces the model to look into other regions of the image for discriminative features where pixel values are non-zero. It is highly likely that the features identified are non-overlapping. Therefore, we are using the sum operation for fusing the features, as the summation would result in a combination of non-overlapping features.
>
> - **Clarity on verification of model’s performance on feature-suppressed datasets:** In the literature, researchers have shown the performance of the existing models by masking different regions of the images. For example, Goswami et al. evaluated the performance of existing face recognition algorithms on face images for different occlusions like eyebrow occlusion, forehead and brow occlusion etc. The performance obtained from these evaluations represented the importance/dependency of the face recognition models on these regions. In the proposed DIVINE, the performance on the feature-suppressed datasets represents the model's performance is dependent on the sparse set of pixels in images.

---

### Review · Reviewer_nW25 · 2023-11-24

**Summary Of Contributions:**

This paper considers the problem of Abridge learning, a subproblem of more general shortcut learning discussed previously in the literature. Abridge learning refers to situations when the model focuses only on a small subset of the most prominent features and as such fails to generalize to OOD situations when these features are not sufficient. The authors propose the DIVINE approach that relies on iteratively training a classifier and masking out the most prominent features detected by the classifier. Finally, DIVINE combines all the retrieved masks to train the final model. The authors check the performance of the method on several standard vision datasets.

**Audience:**

Yes

**Claims And Evidence:**

Yes

**Requested Changes:**

The changes I would like to see are related to the weaknesses listed above:
- Please present more data on the computational complexity of DIVINE
- Please discuss the statistical significance of the results (number of seeds etc.)
- Please explain if this method be used with datasets containing spurious correlations such as Colored MNIST. Of course, experimental evaluation would be best, but a discussion of this point would also be nice.

**Strengths And Weaknesses:**

Strengths:
- The paper considers an important problem, i.e. how to guide models to rely on a diverse set of features rather than any particular strong feature.
- DIVINE, the proposed method, is simple and intuitive.
- The experimental results show that the proposed method works quite well on OOD corrupted data. Additionally, the authors show some interesting analysis and ablation studies, investigating what type of features the network learns and how the $p$ parameter impacts the performance
- The paper is clearly written and easy to follow.


Weaknesses:
- The method seems to be very expensive computationally since it requires additional retraining and computing the Jacobian wrt. to inputs. The authors touch on this in Section 4.4, but I think a more thorough discussion is needed. First of all, the presentation of this issue should be improved -- e.g. although these stats can be inferred from the paper, a table comparing the training time required by the baselines and DIVINE would be nice, and providing total training time on top of the per-epoch time would help the reader. Second, additional data would be useful -- e.g. what's the overhead of the method in each training step? In particular, I'd be curious to find out how slow the Jacobian computation is and how much time it takes to generate the feature-suppressed datasets.
- Checking the performance on corrupted OOD data is interesting by itself, but I think additional experiments would strengthen the paper. For example, how would the method perform on datasets with spurious correlations such as Colored MNIST [1] or domain generalization datasets [2]?
- The authors do not investigate the statistical significance of their results -- in particular, how many seeds were used to train each model? Can you provide the standard deviation?
- The baselines used in the paper are relatively old. It would be great to compare DIVINE to newer methods such as Chefer et al. 2022.

Minor comments:
- The first sentence of the abstract says: "Deep learning algorithms aim to minimize overall classification error". I would say this is an oversimplification as not all deep learning tasks deal with classification
- "We provide a detailed analysis of its trend in the supplementary." - but these results are in Section 4.3 an there does not seem to be any supplementary materials, am I correct?

[1] Arjovsky, M., Bottou, L., Gulrajani, I., & Lopez-Paz, D. (2019). Invariant risk minimization. arXiv preprint arXiv:1907.02893. \
[2] Gulrajani, I., & Lopez-Paz, D. (2020). In search of lost domain generalization. arXiv preprint arXiv:2007.01434.

---

> ### Author Response · Authors · 2023-12-10
> **Detailed response to comments of Reviewer nW25**
>
> We thank the reviewer for providing their feedback and suggestions. Our responses to the reviewer’s questions are provided below:
>
> - **Understanding the overhead of the proposed DIVINE algorithm:** We agree with the reviewer that the proposed DIVINE algorithm is computationally expensive which is also a limitation. As suggested by the reviewer, a thorough discussion is included  in section 4.4 of the updated manuscript. We also calculate the training time of the proposed algorithm on the CIFAR10 dataset and in Table 6 of the revised manuscript. We observe that the overhead training time is 104 seconds for each epoch, which is mostly spent on the computation of Jacobians. This training overhead is discussed in section 4.4 of the updated manuscript.
>
> - **Discussion on Colored MNIST dataset:** The proposed method DIVINE identifies the dominant feature learned by the model and further suppresses it to enforce the model to learn other features. The feature in the paper is represented as the combination of pixels and the proposed method DIVINE sets the identified combination of pixels (feature) to 0, which is a semantic change in the image space. In case of colored MNIST, the spurious correlation is due to color attribute of the digits, which is not a semantic feature. Therefore, the proposed method DIVINE is not applicable to those abridge learning problems where the dominant features are not semantic.
>
> - **Experiments with multiple seeds:** We have computed the standard deviation on the CIFAR10 dataset corresponding to Abridge Learning and the DIVINE method for 3 different seeds. We report a standard deviation of 1.47 for DIVINE and 1.25 for Abridge Learning. Due to time and resource constraints, we will include the standard deviation on other datasets in the camera-ready version of the paper.
>
> - **Comparison with Chefer et el.:** A direct comparison between the proposed DIVINE method and the work from Chefer et al. is not possible due to no intersection in datasets. Furthermore, the technique proposed by Chefer et al. suppresses background features by incorporating a mask to promote the foreground features. Whereas the proposed DIVINE method is generalized and independent of the foreground and background feature separation.
>
> - **Minor Updates in Abstract and typos:** We thank the reviewer for pointing out the typos. We have fixed the typos and updated the abstract accordingly.

---

### Review · Reviewer_cX8r · 2023-11-25

**Summary Of Contributions:**

This paper proposes DIVINE, which is used to make a model more robust to out-of-distribution corruptions to inputs. The method is based off of suppressing input features that have high relative impact on model classification according to the Jacobian. These suppression masks are created in multiple rounds to create multiple versions of feature-suppressed data. A "unified" model is finally trained over the combination of suppressed data and a regularization term based off the suppression masks to get DIVINE. The evaluation is performed over MNIST, CIFAR10, CIFAR100, and TinyImageNet with corruptions.

**Audience:**

Yes

**Claims And Evidence:**

Yes

**Requested Changes:**

Critical:
* Please add scaffolding across the document to summarize key results and ideas before going into details. Fix all writing issues in weaknesses. Ideally, move specific details (e.g., hyperparameters) to an appendix. It should be clear to the reader what each Figure and Section are about without reading the entire paper.

Strengthen:
* Add full ImageNet results.

**Strengths And Weaknesses:**

Strengths:
* The method is intuitive and seems simple to implement.
* Evaluations show improvements over baselines.

Weaknesses:
* The writing is hard to follow. There is no summary of results in the abstract, so the reader does not know why they should read the paper. The method of DIVINE in Section 3 is spending 11 equations discussing DIVINE, but these equations weren't much more helpful than reading Figure 2 and cause high reader fatigue. Evaluation has many details that are best left to an appendix.
* Figures aren't self-contained. Table 1, for example, isn't highlighting results/conclusions, nor even describing the columns. Captions are not capable of being interpreted without referencing the text. Captions do not have a conclusion.
* Spacing/formatting is suboptimal. Page 7, for example, has equations, text, and figures with very different context. Figure 4 is introduced on page 7 but is mentioned in text in page 10.
* Figure 7 seems similar to me for the 3 methods with respect to the feature diversity. Can you explain what the reader should be looking at?
* Table 2 and 3 seem like the same table with different input datasets and different metrics, but this is not obvious to the reader. These tables and their captions are not easy to read or interpret.
* The experiments limit application of the method to few real-world scenarios (e.g., beyond MNIST/CIFAR10). TinyImageNet is a welcome addition, but experiments on the standard ImageNet dataset are more rigorous. Experiments also seem ad-hoc.

Nits:
* Dot for multiplication in Equation 2/3 can be removed. Dot is not used consistently in other equations (e.g., Equation 9) and it looks worse with dot.
* Quotes for "out-of-distribution" in abstract are unnecessary. There is also an extra quote at the end.
* Equation 1 is using X for a dataset of inputs, x, and labels, y. Use a different symbol for dataset, like D, to avoid confusion with inputs.

---

> ### Author Response · Authors · 2023-12-10
> **Detailed response to comments of Reviewer cX8r**
>
> We thank the reviewer for providing their feedback and suggestions. Our responses to the reviewer’s questions are provided below:
>
> - **Adding Summary of Results to Abstract and Moving Evaluation Details to Supplementary:** We have added the summary of the results in the abstract and updated the paper for better readability. We have moved some of the evaluation details to supplementary.
> Improving the captions of figures and tables: We have updated the captions of figures and tables in the revised manuscript for better readability.
>
> - **Improving caption for Figure 7:** Consider the first row in Figure 7. It is evident that the Jacobian regularization and random suppression methods are sensitive to the small region (dominant feature) of the image and fail to focus on the center of the image, where relevant information is present. Most of the region is flat and represented with gray in the Jacobian regularization and random suppression images. However, this is not the case with the unified model, where it is sensitive to a larger region spread out in the image.
>
> - **Updating Tables 2 and 3:** We have updated the captions for Table 2 and Table 3 for better readability.
>
> - **Additional Experiments to showcase the applicability of DIVINE in real-world scenarios:** Due to time and resource constraints, the experimental results of the ImageNet dataset are currently under computation. We shall include the results in the camera-ready version of the manuscript. Additionally, we have performed the experiments on CIAR10-P, CIFAR100-P, and TinyImageNet-P datasets, and the corresponding results are included in section 4.2 of the updated manuscript. We have reported the Flip Rate (FR) and the Overall mean flip rate (Overall mFR) on the CIFAR10-P, CIFAR100-P, and TineImageNet-P datasets. The results are shown in Tables 4 and 5 of the revised manuscript. The proposed unified model outperforms abridge learning on the perturbed datasets. The proposed unified model gives an overall mFR of 5.36%, 3.10%, and 21.85%, corresponding to the CIFAR10-P, CIFAR100-P, and TinyImageNet-P datasets, respectively. On comparing the performance of individual perturbations on all datasets, the proposed unified model trained with the DIVINE method outperforms abridge learning.
>
> - **Improvements in abstract, equations 2 and 3, and using variable for denoting dataset:** We would like to thank the reviewer for pointing it out. We have updated the abstract and the paper with the suggested changes and further removed the dot from equations 2 and 3 to make it consistent with other equations. We would like to highlight that we have used $D$ to represent the Dominant Set in the paper instead of $X$. In the literature, $X$ is commonly used to represent the datasets utilized for training.

---

### Review · Reviewer_SfeS · 2023-11-25

**Summary Of Contributions:**

This paper studied a problem called "Abridge Learning", which was defined as "a sub-problem of Shortcut Learning that deals only with the problem where the model picks up only dominant cues and ignores other relevant features from the input data."
The author proposed a method called "Diverse-Inconspicuous Feature Learning" for this problem which "removes the shortcuts and learning a diverse set of input features."
This method was evaluated on the MNIST, CIFAR-10, CIFAR-100, and TinyImageNet datasets.

**Audience:**

Yes

**Claims And Evidence:**

No

**Requested Changes:**

Minor issues:
- "Abridged Learning" was sometimes used, which I assume was a typo.

**Strengths And Weaknesses:**

Strengths:
- Some people might be interested in masking the input to improve the generalization, and this paper showed a potential direction.

Weaknesses:
- The definition of the so-called "abridge learning" was very unclear. There was no clear mathematical definition of this problem, which prevented people from following this research direction.
- The methodology seems ad hoc and lacks theoretical support.
- The proposed method is limited to visual features. Being $0$ is regarded as an absence of a "feature". The experiments only used image datasets.
- The math writing needs to be improved. For example:
  - The definition of "dataset" $\mathbf{X}$ was not given.
  - Eq. (1) is not a loss function but an optimal parameter.
  - Eq. (2) is not a Jacobian but the Taylor expansion.
  - "$f(x; \theta)$ outputs the probability vector," but the Jacobians were computed "with respect to the true class only." The author used the same notation for both.
- No discussion on the limitations of this work.

---

> ### Author Response · Authors · 2023-12-10
> **Detailed response to comments of Reviewer SfeS**
>
> We thank the reviewer for providing their feedback and suggestions. Our response to the reviewer’s questions are provided below:
>
> - **Definition of Abride Learning:** As mentioned in the paper, Abridge learning is a learning process in which a model learns only the dominant input features while failing to learn other useful input features for the target task. In other words, the model only learns a subset of features enough to achieve promising performance on the given task even if the learnt features are irrelevant for generalization while failing to learn other important features. For example, consider a dataset having images of moon and star patterns, where the moon in the images is situated at the top left, and the stars in the images are situated at the bottom right. Now, for a task to classify the moon vs. star images, as shown in the literature [1], the deep models learn to associate object location with a category, which is an irrelevant feature, illustrating the abridge learning. For more clarity for the reader, we have updated the definition of Abridged Learning in section 1 of the updated manuscript.
>
> - **Theoretical support for DIVINE method:** We would like to mention that the proposed method DIVINE in the paper is one of a kind, which identifies the inconspicuous discriminative input features and uses them to learn a diverse unified model that is generalizable. To support the idea of the proposed method DIVINE, we have provided the mathematical background equations, with theoretical and empirical explanations along with the experimental results on the multiple datasets. We have also provided the visualizations for qualitative analysis of the proposed DIVINE algorithm.
>
> - **Experiments on Image data:** We would like to mention that the scope of this paper is currently limited to image datasets. However, the proposed method is generalizable and can be extended to other modalities like texts and audio signals as well.
>
> - **Considered 0 pixel as the absence of features:** In the literature, there are several papers that consider the 0 pixel value to be the absence of the feature [2,3]. For example, the black image or the black background are considered to be absent from the feature.
>
> - **Minor improvements in mathematical writing:** We have updated the manuscript with the suggested fixes in sections 3 and 3.1 of the revised manuscript.
>
> - **Limitations of the proposed method:** We highlight the following limitations of our work:
> 1. Though the proposed method mitigates Abridged Learning, it is computationally expensive as it utilizes multiple iterations to identify inconspicuous and diverse features and training of the models on the same data.
> 2. The proposed idea is validated only for the image modality in the classification setting. However, the proposed approach can be extended for more modalities and different tasks.
> 3. The proposed method DIVINE is not applicable to those abridge learning problems where the dominant features are not semantic. For example, in the ColoredMNIST dataset, the spurious correlation is due to the color attribute of the digits, which is not a semantic feature, and DIVINE is capable of only highlighting the semantic features in the input data.
>
> We have incorporated the limitations of our work in section 5 of the revised manuscript.
>
> References:
>
> [1] Geirhos, R., Jacobsen, J. H., Michaelis, C., Zemel, R., Brendel, W., Bethge, M., & Wichmann, F. A. (2020). Shortcut learning in deep neural networks. Nature Machine Intelligence, 2(11), 665-673.
>
> [2] Goswami, G., Ratha, N., Agarwal, A., Singh, R., & Vatsa, M. (2018, April). Unravelling robustness of deep learning based face recognition against adversarial attacks. In Proceedings of the AAAI Conference on Artificial Intelligence (Vol. 32, No. 1).
>
> [3] Luo, T., Cai, T., Zhang, M., Chen, S., & Wang, L. (2020). Random mask: Towards robust convolutional neural networks. arXiv preprint arXiv:2007.14249.

---

### Review · Reviewer_XMTZ · 2023-11-26

**Summary Of Contributions:**

This paper studies the shortcut learning, and proposes the Diverse and Inconspicuous Learning (DIVINE) method to maximize learning from a given set of inputs and learning as many discriminative features as possible. Empirical results demonstrate its better generalizability and robustness.

**Audience:**

Yes

**Claims And Evidence:**

Yes

**Requested Changes:**

Please refer to Weaknesses.

**Strengths And Weaknesses:**

Strengths
- The problem investigated by this paper is very important.
- This paper is well-written and organizes well.

Weaknesses
- The abrige learning setting also seems to be suitable for OOD detection/generalization benchmarks. The related experiments should be included here.
-  For Eq. (2), it is a common tayler expansion, while not a "Jacobian". In my view, the Jacobian is just a special matrix.
- There exist some typos, making the readers hard to understand. For example, the Eq. (8) misses a "+". Some figures in this paper are rough.
- It is not clear how to decide the number of feature maps.

---

> ### Author Response · Authors · 2023-12-10
> **Detailed response to comments of Reviewer XMTZ**
>
> We thank the reviewer for providing their feedback and suggestions. Our response to the reviewer’s questions are provided below:
>
> - **Abridged Learning for OOD detection/generalization:** We would like to mention that we have already performed the experiments on OOD (corrupted) datasets and the results are shown in the main paper. We evaluated the unified model on corrupted datasets i.e., CIFAR10-C, CIFAR100-C, and TinyImageNet-C to showcase the generalizability of the unified model against abridge learning.
>
> - **Fixing Equations 2 and 8:** We have incorporated the feedback in section 3.1 (for equation 2) and fixed equation 8 of the revised paper to improve clarity and readability.
>
> - **Number of feature maps:**  In order to decide the number of features, we have computed the running average of the classification accuracy obtained on the original and feature-suppressed dataset using the AL method. We decided to iterate computing feature maps at most 3 times, given the average running classification should not be below 50 percent of the classification accuracy obtained on the original dataset. We have included the computation of feature maps in section 3.6 in the revised manuscript.

---

### Decision · Action_Editor_QbFD · 2023-12-29

**Recommendation:** Reject

**Comment:**

The reviewers' opinions are somewhat divided. However, it is noteworthy that they unanimously agree that the paper discusses an important issue and the proposed method is rational and straightforward. Yet, due to the lack of more comprehensive and in-depth theoretical or experimental analysis, the conclusions of the paper are not sufficiently validated. Therefore, we recommend the authors consider a major revision and resubmission to further refine the paper.

**Audience:**

It is appreciated that the authors have discussed the limitations of their method. However, the need for multiple iterations and model retraining in DIVINE imposes a significant resource burden, which could potentially limit its usage. Furthermore, DIVINE's ineffectiveness in addressing non-semantic dominant features raises questions about its utility in diverse real-world scenarios, potentially reducing its appeal to a broader audience.

**Claims And Evidence:**

Existing research has shown that models often take shortcuts by learning only the dominant input features necessary for confident classification. In this paper, the authors refer to this issue as 'Abridge Learning' and introduce a method called DIVINE to address it. They evaluated their algorithm using several benchmark datasets, including MNIST, CIFAR10, CIFAR100, TinyImageNet, and their variants.

However, as the authors themselves note, the problem of 'Abridge Learning' has been previously studied, with some related works conducting even more comprehensive evaluations. For instance, Li et al. (2023) proposed 'Last Layer Ensemble' as a novel approach for mitigating multiple shortcuts. They assessed their method using two newly created datasets and included a wide range of baseline methods in their comparative experiments. This level of thorough evaluation is missing in the current paper, which weakens the evidence supporting the authors' claims.

**Resubmission Of Major Revision:**

The authors may consider submitting a major revision at a later time.